# Knowledge, Emotions and Stressors in Front-Line Healthcare Workers during the COVID-19 Outbreak in Mexico

**DOI:** 10.3390/ijerph18115622

**Published:** 2021-05-25

**Authors:** Yazmín Hernández-Díaz, Alma Delia Genis-Mendoza, Ana Fresán, Thelma Beatriz González-Castro, Carlos Alfonso Tovilla-Zárate, Isela Esther Juárez-Rojop, María Lilia López-Narváez, José Jaime Martínez-Magaña, Humberto Nicolini

**Affiliations:** 1División Académica Multidisciplinaria de Jalpa de Méndez, Universidad Juárez Autónoma de Tabasco, Jalpa de Méndez, Tabasco 86205, Mexico; yazmin.hdez.diaz@gmail.com (Y.H.-D.); thelma.glez.castro@gmail.com (T.B.G.-C.); 2Instituto Nacional de Medicina Genómica, Servicios de Atención Psiquiátrica, Secretaría de Salud, Ciudad de México 14610, Mexico; adgenis@inmegen.gob.mx; 3Subdirección de Investigaciones Clínicas, Instituto Nacional de Psiquiatría Ramón de la Fuente Muñíz, Ciudad de México 14370, Mexico; fresan@imp.edu.mx; 4División Académica Multidisciplinaria de Comalcalco, Universidad Juárez Autónoma de Tabasco, Comalcalco, Tabasco 86650, Mexico; 5División Académica de Ciencias de la Salud, Universidad Juárez Autónoma de Tabasco, Villahermosa, Tabasco 86150, Mexico; iselajuarezrojop@hotmail.com; 6Hospital General de Yajalón “Dr. José Manuel Velasco Siles”, Secretaría de Salud, Yajalón, Chiapas 29932, Mexico; dralilialonar@yahoo.com.mx; 7Laboratorio de Enfermedades Psiquiátricas, Neurodegenerativas y Adicciones, Instituto Nacional de Medicina Genómica, Ciudad de México 14610, Mexico; jimy.10.06@gmail.com

**Keywords:** COVID-19, SARS-CoV-2, healthcare workers, Mexico, stress

## Abstract

The aim of this study was to explore the knowledge, emotions and perceived stressors by healthcare workers who were in contact with infected patients during the COVID-19 outbreak. An online cross-sectional survey was applied. Data were collected from N = 263 healthcare workers in Tabasco State, Mexico. We developed and administered a questionnaire, which consisted of sociodemographic characteristics, plus four sections. The sections evaluated were (1) knowledge of COVID-19; (2) feelings/emotions during the COVID-19 outbreak; (3) factors that caused stress and (4) factors that helped to reduce stress. Surveyed individuals were divided into three groups: physicians, nurses and other healthcare workers. When we evaluated their knowledge of COVID-19 we observed that the majority of healthcare workers in the three groups reported that they knew about COVID-19. Physicians indicated that they felt insecure about practicing their profession (62.5%) due to the high risk of being in contact with SARS-CoV-2. With regards to stressor factors, the risk of transmitting COVID-19 to their families was the main factor causing moderate to high stress (95.4%). Finally, we found that “your profession puts your life at risk” was the only factor associated with feeling nervous and scared (PR: 3.15; 95% CI: 1.54–6.43). We recommended health education campaigns, introductory courses on COVID-19 and other infectious diseases, management protocols and the provision of protection equipment to health workers in order to reduce personal and professional fears of contagion and to improve the health system in Mexico when facing epidemics.

## 1. Introduction

In December 2019, several cases of pneumonia of unknown etiology were reported in China. The causative agent was identified in January 2020 as Severe Acute Respiratory Syndrome Coronavirus 2 (SARS-CoV-2), while the disease was named COVID-19. In January 2020, COVID-19 was declared a public health emergency of international concern, posing a high risk to countries with vulnerable or deficient health systems [1,2].

The World Health Organization (WHO) indicated that the COVID-19 pandemic had caused over 2, 738, 876 deaths and nearly 124, 535, 520 confirmed cases worldwide as of March 2021 [3]. Country governments adopted preventive and containment measures, and the general population was instructed to take rigorous personal preventive measures [4,5].

In Mexico, the COVID-19 outbreak was confirmed in March 2020, and the government declared a national health emergency. The Mexican government, in coordination with the “Secretaría de Salud” (Mexican Ministry of Health), promptly implemented a series of preventive actions in the country, including the prohibition of mass gathering events and the recommendation of home confinement [6].

This infection is transmitted person-to-person through large droplets spread when infected individuals cough or sneeze. People infected might be asymptomatic, and individuals from all ages are susceptible [7,8]. Evidence suggests that the SARS-CoV-2 transmission starts during the asymptomatic incubation period, which has been estimated to be between two and ten days [7,9].

The clinical features of COVID-19 vary from an asymptomatic state to an acute respiratory distress syndrome and multiorgan dysfunction [10,11]. Individuals with multiple comorbidities are prone to severe infection, and by the end of the first week of the disease, it could progress into pneumonia, respiratory failure and death. There are similarities between symptoms of COVID-19 and earlier beta-coronavirus, as well as unique clinical features of COVID-19 [12,13]. One of the greatest concerns about SARS-CoV-2 is its transmission to healthcare workers who directly care for people affected with this virus, and it is important to protect them.

Healthcare workers are exposed to several job stressors that can adversely affect both their mental and physical health. After the COVID-19 outbreak, healthcare workers were challenged by working in a new context, exhaustion due to heavy workloads, fear of becoming infected and infecting others, stigmatization, understaffing and uncertainty [14,15].

Therefore, we considered that it was important to evaluate different characteristics that affect healthcare workers, as they have experienced varied health problems during the COVID-19 pandemic. Accurate knowledge about COVID-19 is critical for healthcare workers to make appropriate, evidence-based healthcare choices.

To our knowledge, there is no information on the stressors and mental health of healthcare workers concerning COVID-19 in Mexico. Therefore, the present study evaluated healthcare workers who were in direct contact with infected individuals in order to document their satisfaction about their COVID-19 knowledge, the intensity of their perceived work-associated stressors and stress reduction factors, as well as their feelings/emotions during the COVID-19 outbreak in Tabasco State, Mexico. We hypothesized that most healthcare workers would show insecurity and stress caused by COVID-19.

## 2. Material and Methods

### 2.1. Research Design

A cross-sectional survey was performed in order to evaluate healthcare workers in Tabasco State, Mexico. This study is reported following the Strengthening the Reporting of Observational Studies in Epidemiology (STROBE) [16] and criteria of the EQUATOR network. The STROBE checklist was used as a strict guideline for this article (see Appendix A). Ethical approval was granted by the Research Ethics Committee of the Juarez Autonomous University of Tabasco (103/CIP-DACS/2020) in Mexico. Completing the online survey implied informed voluntary consent. Individuals who participated in this study did not receive any financial compensation.

### 2.2. Study Participants and Setting

This study was performed using a sample of health professionals who voluntarily answered the online questionnaire during the period in which it was active. They had to meet the following inclusion criteria: (1) individuals ≥18 years of age, (2) have a degree in medicine, nursing or other healthcare-related profession, (3) work as professionals in care services that facilitated direct contact with patients in any health area and (4) work at any hospital during the COVID-19 outbreak in Tabasco State, Mexico. Professionals working in healthcare administration management or other services where there was no direct contact with patients were excluded, as were students.

A questionnaire was specifically created for this study. Given the social distancing measures during lockdown, the participants were invited to complete and submit the questionnaire online. We used a snowball and referral methodology comprising three parts. Initially, an invitation was emailed to individuals within the authors’ personal networks, requesting them to forward the message to other people who might be interested. Then, we leveraged social media with a Facebook post with a link to the survey. Finally, we forwarded the invitation through WhatsApp to the authors’ personal networks with an invitation to share it forward.

The survey was conducted using the SurveyMonkey^®^ software (SurveyMonkey, San Mateo, CA, USA) to facilitate archiving and data processing. The nature of the study was explained to the participants at the beginning of the electronic interview. Anonymity was carefully preserved.

### 2.3. Data Collection

A self-administered questionnaire was developed by the authors after reviewing up-to-date literature and consulting similar publications. The questionnaire was checked and validated for content and relevance by the authors and external researchers. Considering that in Mexico the first coronavirus (COVID-19) case was confirmed in late February 2020, and it was declared a national health emergency in March, this survey collected information during the early stages of the epidemic, from 1 to 30 April 2020.

The survey consisted of a semi-structured questionnaire with a section inquiring about sociodemographic characteristics, followed by four sections that enquired about (1) knowledge of COVID-19; (2) feelings/emotions during the COVID-19 outbreak; (3) factors that cause stress and (4) factors that help to reduce stress. The participants’ knowledge of COVID-19 was assessed by asking about symptoms of the disease, diagnostic methods and treatment, transmission routes and information about the prevention of COVID-19. In the sections of knowledge and feelings/emotions during the COVID-19 outbreak, the answers were “yes” or “no”. In the section of factors that cause stress, the answers were coded in a Likert scale ranging from “it does not cause me stress” to “it causes me moderate stress” to “it causes me severe stress”. In the section relating to factors that help to reduce stress, the answers were coded in a Likert scale as well, ranging from “it does not reduce my stress” to “it reduces my stress moderately” to “it reduces my stress greatly”.

### 2.4. Data Analysis

All the analyses were conducted using the statistical package IBM SPSS Statistics software (SPSS 23.0). The answers were analyzed using frequencies and percentages for categorical variables. In an initial comparison, the sample was grouped according to the area of expertise: physicians, nurses and other healthcare workers. We examined stressor factors among the three groups and compared the variables by using Chi square (χ^2^) tests.

Finally, to assess the factors associated with stress factors, as well as to predict factors that cause positive attitudes, we calculated crude and adjusted prevalence ratios (PRs) and their respective 95% confidence intervals (95% CIs) using a Poisson loglinear function. The statistical significance was set as *p* ≤ 0.05. Variables such as feeling nervous/stress or feeling positive were the outcome variables. Individuals who answered moderately and very affectively were considered to have a positive attitude towards coronavirus disease.

## 3. Results

### 3.1. Characteristics of the Participants

A total of 398 individuals answered the survey, but only 367 completed all the phases. Following our exclusion criteria, 28 individuals who did not follow the instructions were excluded, and 76 more individuals were excluded because they were students. Then, 263 healthcare workers were included; the majority of them (76%, *n* = 200) were physicians, whereas 15.9% (*n* = 42) were nurses and 21 (8.1%) had other healthcare degrees (nutritionists, psychologists and others).

Table 1 shows the socio-demographic characteristics of participants. The majority of participants were female (65%, *n* = 171), and the main age group was between 18 and 29 years old.

### 3.2. Knowledge, Emotions and Stressors

When we evaluated their knowledge of COVID-19, the majority of participants answered that they knew the symptoms, diagnosis, treatment, transmission and precautionary measures of COVID-19. Regarding their feelings/emotions during the COVID-19 outbreak, our comparisons indicated that physicians were feeling highly insecure about practicing their profession, whereas the group of other healthcare workers felt more secure about practicing their professions (62.5% vs. 33.3%, *p* = 0.01) (Table 2).

With regards to factors that cause stress, physicians had moderate to high stress due to the possibility of transmitting SARS-CoV-2 to their families and friends (χ^2^ = 10.88, *p* = 0.004), the lack of a vaccine or treatment (χ^2^ = 8.57, *p* = 0.01) and putting their lives at risk (χ^2^ = 10.95, *p* = 0.004). Finally, no differences were observed between groups (Table 2) when we evaluated factors that help to reduce stress.

### 3.3. Logistic Regression Analysis

Regarding risk factors associated with feeling nervous and scared in Mexican healthcare workers towards COVID-19, we found that “your profession puts your life at risk” was the only factor associated with these feelings (PR: 3.15; 95% CI: 1.54–6.43) (Table 3).

A second Poisson model was performed to find factors associated with positive attitudes in Mexican healthcare workers towards COVID-19; however, no statistical differences were observed (Table 4).

## 4. Discussion

Our aim was to explore the knowledge, emotions and perceived stressors of healthcare workers who were in direct contact with individuals infected with COVID-19 in Tabasco State, Mexico, during the early stages of the COVID-19 outbreak. First, we found that healthcare workers in Tabasco had knowledge about symptoms, diagnosis, transmission routes and precautionary measures of COVID-19. To date, only one study has assessed the level of knowledge of COVID-19 symptoms among older adults in the Mexican population; however, no studies have evaluated this knowledge in healthcare workers [17]. Having reliable information with an effective dissemination and management of this disease is one of the most important measures to control the epidemic [18]. The current treatment is essentially supportive and symptomatic, and prevention is crucial [12,19]. Additionally, measuring and increasing knowledge of COVID-19 risks and transmission in Mexico is of great public health significance.

Second, another main finding was that the majority of physicians working in hospitals felt insecure practicing their profession. Healthcare professionals are directly involved in diagnosis, treatment and care of individuals infected with SARS-CoV-2. Several healthcare workers have been infected with the virus, while some have even died; therefore, the mental well-being of healthcare workers is critical to overcoming this pandemic [20].

For example, in March 2020 J. Lai surveyed healthcare workers in hospitals equipped with fever clinics or wards for individuals with COVID-19 in Wuhan and other regions in China and reported important psychological burden [21]. It is evident that healthcare workers have a higher risk of being exposed to SARS-CoV-2 than does the general population; additionally, these workers feel highly insecure by social stigma. The risks associated with SARS-CoV-2 infection have shown how stigma and discrimination remain serious barriers to health workers employed in emergency services. An all-round management led by health policymakers and planners seems to be the most appropriate approach to face the stigma and discrimination related to SARS-CoV-2 [22]. Our observations underscored the importance of identifying and managing feelings of insecurity in healthcare workers at an early stage of outbreaks through proper interventions.

Third, among the various stressors related to the COVID-19 outbreak, “risk of transmitting SARS-CoV-2 to family and friends”, “lack of a vaccine or treatment” and “your profession puts your life at risk” were the most frequent in our study. Several studies in other countries have shown similar results during the COVID-19 outbreak. In China for example, the medical staff group scored significantly high on “thought of being in danger”, “the possibility of self-illness”, “worrying about family infection”, “poor sleep quality”, “needing psychological guidance” and “worrying about being infected” [23]. While in Singapore and India, healthcare workers from five major hospitals involved in the care of COVID-19 patients showed moderate to severe levels of stress in 42.6% of the participants [24].

Additionally, healthcare workers are performing a highly demanding job as a result of heavy workloads, extended working hours, time-related pressure and job risk, all of which predispose them to a great deal of stress compared to other professions [25,26].

Although the vaccination process in Mexico started in December 2020, our survey gathered information up to 30 April 2020, and we observed that the lack of a vaccine or a curative treatment was considered an additional risk for healthcare workers [21]. The debate around vaccines has been in the spotlight over the last few months. The reasons why people choose not to be vaccinated are complex. A vaccines advisory group from the WHO identified that complacency, inconvenient access to the vaccines and lack of confidence are key reasons underlying hesitancy [27]. However, it is important to achieve high rates of COVID-19 vaccination in healthcare workers, as they will continue to be in contact with individuals infected with SARS-CoV-2. Worrying about the health of their families, being in isolation and feeling inadequately supported by the government, among other things, could also increase the stress in healthcare workers [28]. Therefore, it is important to have appropriate diagnostic/therapeutic protocols to care for admitted and ambulatory individuals with COVID-19 [29].

Finally, we found that “your profession puts your life at risk” was the only risk factor associated with feeling nervous and scared in our sample. In line with this, anxiety and nervousness are common in any epidemic, although their intensity varies, and disease-related worry or insecurity can have a negative effect on disease management [30,31]. Quantitative studies have shown that front-line healthcare providers who are treating individuals with COVID-19 have high levels of anxiety, irritability, stress and burnout. Chatterjee [32] reported in 2020 that doctors had the highest level of anxiety among healthcare workers in India. Both doctors and nurses perceived a greater level of irritability than other workers; however, other healthcare workers were more likely to experience insomnia. Ruiz-Fernández [33] indicated, on the other hand, that physicians had higher compassion fatigue (CF) and burnout (BO) scores, while nurses had higher compassion satisfaction (CS) scores; nonetheless, the perceived stress was similar in both occupations. Professionals working in specific COVID-19 units and in emergency departments had higher CF and BO scores, while their levels of CS and perceived stress were similar regardless of the workplace.

In other countries the private sector has been neglected and left entirely alone to deal with the problem. Most dentists in Poland have followed SARS-CoV-2 recommendation protocols and have restricted their practices to admitting only individuals with pain or incomplete treatment causing a financial impact [34]. In Spain, the COVID-19 pandemic has had economic repercussions as only urgent treatments could be available during the State of Alarm. The level of assistance has also been affected, reducing the number of treated patients, although this quantity has been different in private and public surgeries [35].

Considering the evidence discussed above, we could say that health emergencies, such as the COVID-19 outbreak, highlight the necessity of having preliminary and constant training programs for healthcare workers, in order to address preventive strategies and to better cope during disaster situations and large medical emergencies [36]. Training programs not only would increase professional knowledge and improve practical skills when responding to emergencies, but these strategies could also have a positive impact on healthcare workers’ mental health during medical crises.

Another tool that could help reduce anxiety and stress in healthcare workers is the implementation of psychological support systems. For instance, in February 2020, China announced setting up a nation-wide psychological assistance hotline in order to help their citizens during this pandemic [36,37]. In the rest of the world, however, evaluations and mental health interventions targeting front-line healthcare workers are relatively scarce despite the fact that these workers are at high risk of mental health disorders. Providing telephone-based and internet-based counselling services and platforms with psychological guidance specifically designed for healthcare workers should be a priority. Currently, in Mexico there are only a few mental healthcare programs aimed at helping healthcare workers, some of which are led by the federal government of Mexico and mental health associations [38].

Our work has some limitations that should be taken into consideration. The participants were asked to answer very specific questions that might not have covered the complex interaction of knowledge, emotions and perceived stressors of healthcare workers in Tabasco during the first months of the COVID-19 outbreak in Mexico. Thus, further investigations into the long-term, psychological implications for this population should be performed. To date, no other studies have been performed in Latin-American populations, and this exploratory study could help visualize the panorama of professional healthcare workers who live in different situations than Asian and European health workers. Another limitation was that our study sample was over-represented by physicians; therefore, our findings should be interpreted with caution. A more systematic, inclusive sampling method would improve representativeness and a generalization of the findings.

Additionally, selection bias may exist. The voluntary nature of the survey and data being collected online may have created a selection bias and a lack of validity in the absence of face-to-face interviews. This means that we could not reach healthcare workers with internet connectivity issues; thus, the generalizability of our findings may be limited. Nonetheless, web-based surveys may be the most appropriate method of data collection during an epidemic as this method does not cause SARS-CoV-2 transmission. We also considered that anonymity allowed healthcare workers to feel comfortable answering the questionnaire. It is necessary to add that no power calculations were undertaken prior to this study, as our purpose was only descriptive.

## 5. Conclusions

Our study indicates that the COVID-19 outbreak has caused significant stress, emotional turmoil and concern in healthcare providers, regardless of having up-to-date knowledge of COVID-19, particularly during the first months of the outbreak. Healthcare workers play an important role during epidemics; therefore, listening to their concerns and points of view during a medical crisis is vital for building a responsive and reliable health system.

We recommend health education campaigns and the introduction of essential knowledge and management protocols in order to reduce personal and professional fears of contagion. Although the Mexican government has taken major steps to limit the spread of SARS-CoV-2, it is necessary that health authorities immediately start creating strategies to support healthcare workers. Organizations should develop targeted strategies to mitigate key stressors and improve healthcare workers’ mental well-being.

Longitudinal studies are needed to investigate the psychological impact of the COVID-19 outbreak and the acceptability of a COVID-19 vaccine among healthcare workers in the long term.

## Figures and Tables

**Table 1 ijerph-18-05622-t001:** Characteristic of individuals included in the survey.

Characteristics	*n*	%
*Gender*		
Male	92	35.0
Female	171	65.0
*Age*		
18–29 years	107	40.7
30–39 years	75	28.5
40–49 years	53	20.2
50–59 years	24	9.1
>60 years	4	1.5
*Education*		
Degree in Medicine	130	59.4
Specialty in Medicine	120	45.6
Associated degree	13	4.9
*Marital status*		
Single	142	54.0
Widowed	97	36.9
Separated/divorced	22	8.4
Widowed	2	0.8
*Occupation*		
Medical physician	200	76.0
Nurse	42	16.0
Other (psychologist, nutritionist, social worker, etc.)	21	8.0

**Table 2 ijerph-18-05622-t002:** Summary of the survey findings.

Survey Questions	Response	All Sample *n* = 263 (%)	Physicians *n* = 200 (%)	Nurses *n* = 42 (%)	Other Healthcare Workers *n* = 21 (%)	
*Knowledge of COVID-19*						
Do you have enough information about the symptoms of COVID-19?	Yes	208 (79.1)	164 (82.0)	28 (66.7)	16 (76.2)	χ^2^ = 5.05, *p* = 0.08
Do you have enough information about the diagnosis of COVID-19?	Yes	231 (87.8)	179 (89.5)	34 (81.0)	18 (85.7)	χ^2^ = 2.46, *p* = 0.29
Do you have enough information about the treatment of COVID-19?	Yes	138 (52.5)	110 (55.0)	17 (40.5)	11 (52.4)	χ^2^ = 2.93, *p* = 0.23
Do you have enough information about the transmission routes of COVID-19?	Yes	242 (92.4)	187 (94.0)	36 (85.7)	19 (90.5)	χ^2^ = 3.46, *p* = 0.17
Do you have enough information about the prevention of COVID-19?	Yes	251 (95.4)	192 (96.0)I	39 (92.9)	20 (95.2)	χ^2^ = 1.47, *p* = 0.81
*Feelings/emotions during the COVID-19 outbreak*						
Do you feel nervous and scared?	Yes	175 (66.5)	135 (67.5)	28 (66.7)	12 (57.1)	χ^2^ = 0.91, *p* = 0.63
Do you feel insecure about exercising your profession?	Yes	153 (58.2)	125 (62.5)	21 (50.0)	7 (33.3)	χ^2^ = 8.01 ***p* = 0.01**
Have you thought about not working?	Yes	66 (25.1)	51 (25.5)	8 (19.0)	7 (33.3)	χ^2^ = 1.59, *p* = 0.45
Do you feel sad all day?	Yes	66 (25.1)	51 (25.5)	8 (19.0)	7 (33.3)	χ^2^ = 1.59, *p* = 0.45
If it were optional, would you choose to work in an area where you would not be exposed to viruses?	Yes	183 (69.6)	145 (72.5)	27 (64.3)	11 (52.4)	χ^2^ = 4.29, p = 0.11
*Factors that cause you stress*						
The risk of transmitting COVID-19 to your family and friends	Moderate and high stress	243 (92.4)	190 (95.0)	37 (88.1)	16 (76.2)	χ^2^ = 10.88, ***p* = 0.004**
Lack of a vaccine or treatment	Moderate and high stress	213 (81.0)	167 (83.5)	34 (81.0)	12 (57.1)	χ^2^ = 8.57, ***p* = 0.01**
Your profession puts your life at risk	Moderate and high stress	219 (83.3)	175 (87.5)	30 (71.4)	14 (66.7)	χ^2^ = 10.95, ***p* = 0.004**
Lack of protective measures or equipment in hospitals	Moderate and high stress	257 (97.7)	196 (98.0)	40 (95.2)	21 (100.0)	χ^2^ = 1.72, *p* = 0.43
Strict measures of admission and management of patients in hospitals	Moderate and high stress	202 (76.8)	157 (78.5)	33 (78.6)	12 (57.1)	χ^2^ = 4.95, *p* = 0.08
*Factors that can help reduce stress*						
Positive attitude of healthcare personnel	Moderately and very effective in reducing stress	238 (90.5)	178 (89.0)	40 (95.2)	20 (95.2)	χ^2^ = 12.16, *p* = 0.33
Protective equipment provided to you by the hospital	Moderately and very effective in reducing stress	188 (71.5)	142 (71.0)	31 (73.8)	15 (71.4)	χ^2^ = 0.13, *p* = 0.93
Clear guidelines for the management of infected patients	Moderately and very effective in reducing stress	228 (86.7)	177 (88.5)	34 (81.0)	17 (81.0)	χ^2^ = 2.36 *p* = 0.30
Decrease in the number of positive cases in the country	Moderately and very effective in reducing stress	204 (77.6)	153 (76.5)	35 (83.3)	16 (76.2)	χ^2^ = 0.95, *p* = 0.62
Shorter working days	Moderately and very effective in reducing stress	229 (87.1)	177 (88.5)	35 (83.3)	17 (81.0)	χ^2^ = 1.58, *p* = 0.45

**Table 3 ijerph-18-05622-t003:** Factors associated with feeling nervous and scared of Mexican healthcare workers towards COVID-19.

Factors	β	SE	PR	95% CI	*p*-Value
Gender (male)	0.21	0.16	1.23	0.89–1.71	0.20
Age	0.03	0.07	1.03	0.90–1.19	0.61
Occupation	0.37	0.13	1.03	0.80–1.34	0.77
The risk of transmitting COVID-19 to your family and friends	0.19	0.39	1.21	0.55–2.65	0.62
Lack of a vaccine or treatment	0.20	0.24	1.22	0.75–1.98	0.41
Your profession puts your life at risk	1.14	0.36	3.15	1.54–6.43	**0.002**
Lack of protective measures or equipment in hospitals	0.89	1.01	2.43	0.33–17.87	0.38
Strict measures of admission and management of patients in hospitals	0.10	0.20	1.10	0.73–1.65	0.62

**Table 4 ijerph-18-05622-t004:** Factors associated with positive attitudes of Mexican healthcare workers towards COVID-19 by multiple regression analysis.

Characteristics	β	SE	PR	95% CI	*p*-Value
Gender (male)	0.04	0.13	1.04	0.79–1.36	0.77
Age	0.008	0.06	1.00	0.89–1.13	0.89
Occupation	0.04	0.10	1.04	0.85–1.28	0.65
Protective equipment provided to you by the hospital	0.09	0.16	1.09	0.78–1.52	0.58
Clear guidelines for the management of infected patients	0.21	0.22	1.23	0.78–1.93	0.35
Decrease in the number of positive cases in the country	0.04	0.17	1.04	0.73–1.47	0.82
Shorter working days	0.06	0.20	1.07	0.71–1.60	0.73

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
