# Peer review of "Knowledge, Emotions and Stressors in Front-Line Healthcare Workers during the COVID-19 Outbreak in Mexico"

_ijerph, 2021, doi:10.3390/ijerph18115622_

Round 1

Reviewer 1 Report

Please clarify that you are not talking about all healthcare workers by altering your terminology to direct patient care medical staff or something similar. For example in abstract you say “ The aim of this study was to explore the knowledge, emotions and perceived stressors of healthcare”.  In the intro you say….”The present study evaluated healthcare workers in Tabasco State, Mexico”…..Even the title implies all healthcare workers are included.

In Methods, you need to explain how recruitment for the online questionnaire occurred and discuss the limitations and bias that might be introduced by using this approach

I appreciate that you are trying to understand the factors that influence some of the outcomes you measured but you need to provide more information on how you got to these multivariate models. For example, did you conduct univariate analysis for factors as a first step? Then did you do a step-wise forward or backward process of including the factors significant in univariate analysis into the multivariate model?  In either case  I would expect your final model would include only variables that were significant predictors.  

So the way you report your logistic models is problematic……Logisitic regression is for rare outcomes so in the case of being nervous or scared > 67% of the population reports Yes….it that case you should be using a Poisson loglinear  function because of the high percentage of the population with the outcome.  A number of authors have recommended using Poisson models to estimate the RR due to the increasing differential between the OR and RR when the outcome incidence exceeds 10%

  1. McNutt LA, Wu C, Xue X, Hafner JP. Estimating the Relative Risk in Cohort Studies and Clinical Trials of Common Outcomes. Am J Epidemiol 2003; 157(10):940-3.
  2. Zou G. A Modified Poisson Regression Approach to Prospective Studies with Binary Data. Am J Epidemiol 2004; 159(7):702-6.
  3. Sander Greenland , Model-based Estimation of Relative Risks and Other Epidemiologic Measures in Studies of Common Outcomes in Case-Control Studies, American Journal  of Epidemiology 2004;160:301-305

For the outcome of positive attitude it is confusing as your table reports that over 90% of people reported that having a positive attitude was moderately or very effective in reducing stress but I don’t see where you recorded if people said they had a positive attitude or not….. So, it seems like that for a logistic rare outcome model your outcome should be the risk of NOT believing that a positive attitude is moderately or very effective in reducing stress and your predictors are the same.

Author Response

Reviewer #1

Comment [1]: Please clarify that you are not talking about all healthcare workers by altering your terminology to direct patient care medical staff or something similar. For example in abstract you say “ The aim of this study was to explore the knowledge, emotions and perceived stressors of healthcare”.  In the intro you say….”The present study evaluated healthcare workers in Tabasco State, Mexico”…..Even the title implies all healthcare workers are included.

Reply [1]: Thank you very much for your comment. We agree, we rewrote the sentences to clarify this point (Page 1, Lines 2-3; 26-27; Page 2, 87-91).

Change in the manuscript. Title: Knowledge, Emotions and Stressors in Front-Line Healthcare Workers During the COVID-19 Outbreak in Mexico.

Abstract section. The aim of this study was to explore the knowledge, emotions and perceived stressors by healthcare workers who were in contact with infected patients during the COVID-19 outbreak.

Introduction section. The present study evaluated healthcare workers who were in direct contact with infected individuals in order to document their satisfaction about their COVID-19 knowledge, the intensity of their perceived work-associated stressors and stress reduction factors, as well as their feelings/emotions during the COVID-19 outbreak in Tabasco State, Mexico.

Comment [2]: In Methods, you need to explain how recruitment for the online questionnaire occurred and discuss the limitations and bias that might be introduced by using this approach.

Reply [2]: Thank you for your kindly suggestion. The information was added (Page 3: Lines 114-118, Page 9: Lines 304-309).

Change in the manuscript. Materials and Methods. We used a snowball and referral methodology comprised of three parts. Initially, an invitation was emailed to individuals within the authors’ personal networks, requesting them to forward the message to other people who might be interested. Then, we leveraged social media with a Facebook post with a link to the survey. Finally, we forwarded the invitation through WhatsApp to the authors’ personal networks with an invitation to share it for-ward.

Discussion section. Additionally, selection bias may exist. The voluntary nature of the survey and data being collected online, may have created a selection bias and a lack of validity in the ab-sence of face-to-face interviews, which means that we could not reach healthcare workers with internet connectivity issues; then, the generalizability of our findings may be limited. Nonetheless, web-based surveys may be the most appropriate method of data collection during an epidemic as this method does not cause SARS-CoV-2 transmission.

Comment [3]: I appreciate that you are trying to understand the factors that influence some of the outcomes you measured but you need to provide more information on how you got to these multivariate models. For example, did you conduct univariate analysis for factors as a first step? Then did you do a step-wise forward or backward process of including the factors significant in univariate analysis into the multivariate model?  In either case  I would expect your final model would include only variables that were significant predictors.  So the way you report your logistic models is problematic……Logisitic regression is for rare outcomes so in the case of being nervous or scared > 67% of the population reports Yes….it that case you should be using a Poisson loglinear  function because of the high percentage of the population with the outcome.  A number of authors have recommended using Poisson models to estimate the RR due to the increasing differential between the OR and RR when the outcome incidence exceeds 10%

  1. McNutt LA, Wu C, Xue X, Hafner JP. Estimating the Relative Risk in Cohort Studies and Clinical Trials of Common Outcomes. Am J Epidemiol 2003; 157(10):940-3.
  2. Zou G. A Modified Poisson Regression Approach to Prospective Studies with Binary Data. Am J Epidemiol 2004; 159(7):702-6.
  3. Sander Greenland , Model-based Estimation of Relative Risks and Other Epidemiologic Measures in Studies of Common Outcomes in Case-Control Studies, American Journal  of Epidemiology 2004;160:301-305

For the outcome of positive attitude it is confusing as your table reports that over 90% of people reported that having a positive attitude was moderately or very effective in reducing stress but I don’t see where you recorded if people said they had a positive attitude or not….. So, it seems like that for a logistic rare outcome model your outcome should be the risk of NOT believing that a positive attitude is moderately or very effective in reducing stress and your predictors are the same.

Reply [3]: Thanks very much for the reviewer’s helpful comment. New analyzes were carried out to correct this point (Page 4, Lines 150-155; Page 6, 182-195).

Change in the manuscript. Materials and Methods section. Data analysis. Finally, to assess the factors associated with stress factors as well as to predict factors that cause positive attitudes, we calculated crude and adjusted prevalence ratios (PR) and their respective 95% confidence intervals (95% CI) using a Poisson loglinear function. The statistical significance was set as p≤0.05. Variables such as feeling nervous/stress or feeling positive were the outcome variables. Individuals who answered moderately and very affective were considered as individuals with positive attitude towards coronavirus disease.

Results section. Logistic regression analysis. Regarding risk factors associated with felling nervous and scared in Mexican healthcare workers towards COVID-19, we found that “your profession puts your life at risk” was the only factor associated with feeling nervous and scared (PR:3.15; 95% CI:1.54-6.43) (Table 3). A second Poisson model was performed to find factors associated with positive attitudes in Mexican healthcare workers towards COVID-19; however, no statistical differences were observed (Table 4).

Reviewer 2 Report

Dear authors,

thanks for your contribution proposal.

Minor language issues.

SPSS in an IBM software, please report it correctly.

Author Response

Reviewer #2

Comment [1]: SPSS is an IBM software, please report it correctly.

Reply [1]: Thank you very much for your comment; we have corrected the mistake. (Page 3, Lines 144-145).

Change in the manuscript. Materials and Methods section. All the analyses were conducted using the statistical package IBM SPSS Statistics software (SPSS 23.0).

Reviewer 3 Report

This is an important and interesting study concerning SARS-Cov19 problems in Mexican healthcare. While studies of this kind are on demand, however  topic itself huge and some criticism should be raised:

Abstract:

Should be re-written after accounting all remarks below

Introduction

L58-L90 First part of an Introduction is lengthy and should be cut short

L91-L99 - this part of an Introduction should be partially moved into Discussion section

There is no clear hypothesis, please specify what your goals and study expectations were.

Materials and methods

L129 - Survey Monkey software is missing manufacturers' description

Results

The Hosmer–Lemeshow test has certain limitations. Is there any particular reason why authors chose exactly this one?

Discussion section

Start of the section should be at L229, everything prior to it (L221-228) should be moved farther into the section, where authors should discuss their findings with other publications. Also, there were no mention in between state healthcare and private sector, whereas e.g. most dentists and PTs operate and had serious issues of different origin during first wave of the pandemic; thus there are much more articles form different countries authors may discuss with, for instance https://www.mdpi.com/1660-4601/18/3/1281 but entire special issue scope was healthcare SARS-Cov19 issues and you will find plenty of potential citations. This will surely improve an overall manuscript quality.

Author Response

Reviewer #3

Comment [1]: Abstract: Should be re-written after accounting all remarks below.

Reply [1]: Thank you very much for your comment. We agree, we rewrote the abstract (Page 1, Lines 26-42).

Change in the manuscript. Abstract section. The aim of this study was to explore the knowledge, emotions and perceived stressors by healthcare workers who were in contact with infected patients during the COVID-19 outbreak. An online cross-sectional survey was applied. Data were collected from N = 263 healthcare workers in Tabasco State, Mexico. We developed and administered a questionnaire which consisted of sociodemographic characteristics plus four sections. The sections evaluated were (1) knowledge on COVID-19; (2) feelings/emotions during the COVID-19 outbreak; (3) factors that caused stress and (4) factors that help to reduce stress. Surveyed individuals were divided into three groups: physicians, nurses and other healthcare workers. When we evaluated their knowledge on COVID-19 we observed that the majority of healthcare workers in the three groups reported that they knew about COVID-19. Physicians indicated that they felt insecure about practicing their profession (62.5%) due to a high risk of being in contact with SARS-CoV-2. With regards to stressor factors, the risk of transmitting COVID-19 to their families was the main factor causing moderate to high stress (95.4%). Finally, we found that “your profession puts your life at risk” was the only factor associated to feeling nervous and scared (PR:3.15; 95% CI:1.54-6.43). We recommend health education campaigns, introductory courses on COVID-19 and other infectious diseases, management protocols and to provide protection equipment to health workers in order to reduce personal and professional fears of contagion and to improve the health system in Mexico when facing epidemics.

Comment [2]: Introduction. L58-L90 First part of an Introduction is lengthy and should be cut short. L91-L99 - this part of an Introduction should be partially moved into Discussion section

Reply [2]: Thank you very much for your comment. The information has been re-organized (Page 2, Lines 54-76).

Change in the manuscript. Introduction section. The World Health Organization (WHO) indicated that the COVID-19 pandemic had caused over 2, 738, 876 deaths and nearly 124, 535, 520 confirmed cases worldwide as of March 2021 [3]. Country governments adopted preventive and containment measures and the general population was instructed to take rigorous personal preventive measures [4, 5].

In Mexico, the COVID-19 outbreak was confirmed in March 2020 and the government declared a national health emergency. The Mexican government in coordination with the “Secretaría de Salud” (Mexican Ministry of Health) promptly implemented a series of preventive actions in the country, including the prohibition of massive events and the recommendation of home confinement [6].

This infection is transmitted person-to-person through large droplets spread when infected individuals cough or sneeze; people infected might be asymptomatic and individuals from all ages are susceptible [7, 8]. Evidence suggests that the SARS-CoV-2 transmission starts during the asymptomatic incubation period, which has been estimated to be between two and ten days [7, 9].

The clinical features of COVID-19 vary from an asymptomatic state to an acute respiratory distress syndrome and multi organ dysfunction [10, 11]. Individuals with multiple comorbidities are prone to a severe infection and by the end of the first week of the disease, it could progress into pneumonia, respiratory failure and death. There are similarities between symptoms of COVID-19 and earlier beta-coronavirus, as well as unique clinical features of COVID-19 [12, 13]. One of the greatest concerns about SARS-CoV-2 is its transmission to healthcare workers, who directly care for people affected with this virus, there the importance to protect them.

Comment [3]: Introduction. There is no clear hypothesis, please specify what your goals and study expectations were.

Reply [3]: Thanks very much for the reviewer’s helpful comment. We added the information solicited (Page 2, Lines 91-92).

Change in the manuscript. Introduction section. We hypothesized that most healthcare workers would show insecurity and stress caused by COVID-19.

Comment [4]: Materials and methods. L129 - Survey Monkey software is missing manufacturers' description.

Reply [4]: Thank you. We have corrected the mistake (Page 3, Lines 120-121).

Change in the manuscript. Materials and Methods section. The survey was conducted using the SurveyMonkey® software (SurveyMonkey, San Mateo, CA) to facilitate archiving and data processing.

Comment [5]: Results. The Hosmer–Lemeshow test has certain limitations. Is there any particular reason why authors chose exactly this one?

Reply [5]: Thanks very much for the reviewer’s helpful comment. New analyzes were carried out to correct this point (Page 4, Lines 150-155; Page 6, 182-195).

Change in the manuscript. Materials and Methods section. Data analysis. Finally, to assess the factors associated with stress factors as well as to predict factors that cause positive attitudes, we calculated crude and adjusted prevalence ratios (PR) and their respective 95% confidence intervals (95% CI) using a Poisson loglinear function. The statistical significance was set as p≤0.05. Variables such as feeling nervous/stress or feeling positive were the outcome variables. Individuals who answered moderately and very affective were considered as individuals with positive attitude towards coronavirus disease.

Results section. Logistic regression analysis. Regarding risk factors associated with felling nervous and scared in Mexican healthcare workers towards COVID-19, we found that “your profession puts your life at risk” was the only factor associated with feeling nervous and scared (PR:3.15; 95% CI:1.54-6.43) (Table 3). A second Poisson model was performed to find factors associated with positive attitudes in Mexican healthcare workers towards COVID-19; however, no statistical differences were observed (Table 4).

Comment [6]: Discussion section. Start of the section should be at L229, everything prior to it (L221-228) should be moved farther into the section, where authors should discuss their findings with other publications. Also, there were no mention in between state healthcare and private sector, whereas e.g. most dentists and PTs operate and had serious issues of different origin during first wave of the pandemic; thus there are much more articles form different countries authors may discuss with, for instance https://www.mdpi.com/1660-4601/18/3/1281 but entire special issue scope was healthcare SARS-Cov19 issues and you will find plenty of potential citations. This will surely improve an overall manuscript quality.

Reply [6]: We thank reviewer for this insightful comment. We rewrote this section and we added the information solicited (Page 7, Lines 197-272).

Change in the manuscript. Discussion section. Our aim was to explore the knowledge, emotions and perceived stressors of healthcare workers who were in direct contact with individuals infected with COVID-19 in Tabasco State, Mexico during the early stages of the COVID-19 outbreak. First, we found that healthcare workers in Tabasco had knowledge about symptoms, diagnosis, trans-mission routes and precautionary measures of COVID-19. Up to today, only one study has assessed the level of knowledge of COVID-19 symptoms among older adults in Mexican population; however, no studies have evaluated this knowledge in healthcare workers [17]. Having reliable information with an effective dissemination and management of this disease is one of the most important measures to control the epidemic [18]. The current treatment is essentially supportive and symptomatic, and prevention is crucial [12, 19]. Additionally, measuring and increasing the knowledge on COVID-19 risks and transmis-sion in Mexico is of great public health significance.

Second, another main finding was that the majority of physicians working in hospi-tals felt insecure practicing their profession. Healthcare professionals are directly involved in diagnosis, treatment and care of individuals infected with SARS-CoV-2 and several healthcare workers have been infected with the virus and some have even died; therefore, the mental well-being of healthcare workers is critical to overcoming this pandemic [20].

For instance, J. Lai in March 2020 surveyed heath care workers in hospitals equipped with fever clinics or wards for individuals with COVID-19 in Wuhan and other regions in China and reported important psychological burden [21]. It is evident that healthcare workers have a higher risk of being exposed to SARS-CoV-2 than the general population; additionally, these workers feel highly insecure by social stigma. The risks associated with SARS-CoV-2 infection have shown how stigma and discrimination remain serious barri-ers to health workers employed in emergency services. An all-round management lead by health policymakers and planners seems to be the most appropriate approach to face the stigma and discrimination related to SARS-CoV-2 [22]. Our observations underscored the importance of identifying and managing insecurity feelings in healthcare workers at an early stage of outbreaks through a proper intervention.

Third, among the various stressors related to the COVID-19 outbreak, “risk of trans-mitting SARS-CoV-2 to family and friends”, “lack of a vaccine or treatment” and “your profession puts your life at risk” were the most frequent in our study. Several studies in other countries have shown similar results during the COVID-19 outbreak. In China for example, the medical staff group scored significantly high on "thought of being in dan-ger", "the possibility of self-illness", "worrying about family infection", "poor sleep quality", "needing psychological guidance" and "worrying about being infected" [23]. While in Singapore and India, healthcare workers from 5 major hospitals involved in the care of COVID-19 patients, showed moderate to severe levels of stress in 42.6% of the participants [24].

Additionally, healthcare workers are performing a highly demanding job as a result of heavy workloads, extended working hours, time-related pressure and job risk, which predispose them to a great deal of stress compared to other professions [25, 26].

Although the vaccination process in Mexico started in December 2020, our survey gathered information up to the 30th of April 2020 and we observed that the lack of a vac-cine or a curative treatment was considered an additional risk for healthcare workers [21]. The debate around vaccines has been in the spotlight over the last few months. The rea-sons why people choose not to be vaccinated are complex. A vaccines advisory group from the WHO identified that complacency, inconvenience access to the vaccines and lack of confidence are key reasons underlying hesitancy [27]. However, it is important to achieve high rates of COVID-19 vaccination in healthcare workers, as they will continue to be in contact with individuals infected with SARS-CoV-2. Worrying about their family health, being in isolation, and feeling inadequately supported by the government among other thing, could also increase the stress in healthcare workers [28]. Therefore, it is im-portant to have appropriate diagnostic/therapeutic protocols to care for admitted and ambulatory individuals with COVID-19 [29].

Finally, we found that “your profession puts your life at risk” was the only risk factor associated with feeling nervous and scared in our sample. In line with this, anxiety and nervousness are common in any epidemic; although their intensity varies, disease-related worry or insecurity can have a negative effect on disease management [30, 31]. Quantita-tive studies have shown that front-line healthcare providers who are treating individuals with COVID-19 have high levels of anxiety, irritability, stress and burnout. Chatterjee [32] reported in 2020 that doctors had the highest level of anxiety among healthcare workers in India. Both, doctors and nurses perceived a greater level of irritability than other workers; other healthcare workers however, were more likely to experience insomnia. Ruiz-Fernández [33] indicated on the other hand, that physicians had higher compassion fatigue (CF) and burnout (BO) scores, while nurses had higher compassion satisfaction (CS) scores; nonetheless, the perceived stress was similar in both occupations. Profession-als working in specific COVID-19 units and in emergency departments had higher CF and BO scores, while their levels of CS and perceived stress were similar regardless of the workplace.

In other countries the private sector has been neglected and left entirely alone to deal with the problem. Most dentists in Poland have followed SARS-CoV-2 recommendation protocols and have restricted their practices to admitting only individuals with pain or incomplete treatment causing a financial impact [34]. In Spain, the COVID-19 pandemic has had economic repercussions as only urgent treatments could be available during the State of Alarm. The level of assistance has also been affected, reducing the number of treated patients, although this quantity has been different in private and public surgeries [35].

Round 2

Reviewer 3 Report

All of my remarks were successfully addressed. I am satisfied now.

This manuscript is a resubmission of an earlier submission. The following is a list of the peer review reports and author responses from that submission.

Round 1

Reviewer 1 Report

ijerph-1154119

Healthcare Workers Knowledge, Emotions and Stressors During the COVID-19 Outbreak in Mexico.

The researchers missed an opportunity to understand in more detail the work related stressors faced by hospital workers in all jobs during the pandemic.  To keep the questionnaire short, open ended comments could have been allowed or branching questions for more details when concerns were raised. For example, more information about what were the missing protective measures and equipment experiences? What were the guidelines for treatment provided?  The questions asked and data analysis provided to not add much to the literature.  We already know that the vast majority of hospital workers were severely stressed by their working conditions during this time….what have you added?  What is the benefit of looking as physicians vs nurses vs other health care workers.

On line 66 re-write “Accurate knowledge on COVID-19 66 is one critical point to make appropriate evidence-based health care choices.” To…..Accurate knowledge about Covid-19 is critical so that healthcare workers can make appropriate evidence-based health care choices.

On line 68 re-write “The present  study was conducted in order to explore the level of knowledge, emotions and perceived stressors of healthcare workers… to The present study was conducted among health care workers in Tabasco state, Mexico in order to document their satisfaction with their knowledge about Covid-19, the intensity of their perceived work-associated stressors and stress reduction factors and their feelings/emotions during the Covid-19 outbreak.  

The background is missing reference to the many papers on work-related stress and burnout among healthcare workers during the pandemic.

Line 74/75We need references and explanation of the Equator network and Strobe data

Line 81. What do you mean by every user of social media….in Mexico?  In Tabasco state?  What social media?  How were they contacted? This is confusing when combined with section 2.3 that says health care workers who were at hospitals in Tabasco state  were eligible….please combine 2.2 and 2.3 and give us a clearer idea of who you reached out to and how were they invited to participate in the survey?

Why were health care workers without a degree in nursing, medicine or “career related to health care” excluded.  Aides, orderlies, cleaners, maintenance, kitchen, clerical staff were also in a high risk environment, why were they excluded?

What is meant by working at any Mexican public health service? Above you say at hospitals, is this different?

Hopefully you collected data on number hours per day and days per week worked over some period like last month etc. to add to Table 1.

Line 117 you say you evaluated their knowledge about covid-19 but it seems that you evaluated their perception of whether they had been provided sufficient information about covid-19 not whether they actually knew the symptoms, diagnosis, treatment, transmission routes or prevention of covid 19.  This is very different.

Line 119 and table 2 you asked about feelings/emotions during the covid 19 outbreak. Please define the timing of the questionnaire relative to the outbreak in this region…what was the caseload, positivity rate relative to the first case reports from China and Mexico.

How were the questions about feelings/emotions framed?  Did you ever, do you now?  Frequency? Intensity?

Seems unfortunate that you did not ask about what health symptoms of stress the respondents were experiencing.

It seems that the main interest is in comparing personnel rather than in comparing which stressors and stress reduction factors are more important?  Strongly suggest you look at the sub groups of moderate and high or moderate and very effective to differentiate which factors are more important.  

Note that the N for physicians is 10x other health care workers n and 5x nurse n….researchers should discuss the strength of their findings relative to N.

A more nuanced look at what factors track together may have provided some novelty to this work.

Reviewer 2 Report

Dear Authors,

the impact of the current pandemic is transversal to human nature itself, affecting health and well-being at all levels.

Your contribution proposal needs a finishing touch and some insights in order to deserve publication.

  • Introduction: contextualize the pandemic scenario underway, with clear references to the international (WHO), supranational and national bodies involved in the management of the emergency
  • you're writing for a scientific journal, please reformulate all the text
  • why not SARS-CoV-2 among keywords?
  • [Materials and Methods] which limitations in order of cross-sectional study? which limitations in order of a web-based survey? which biases?
  • [Materials and Methods] the questionnaire administered is tailor-made, right? Does it contain parts of validated questionnaires? Please provide detailed informations
  • [Materials and Methods] which software for data analysis
  • [Results] this section is too poor;
  • what about risk perception?
  • deal with gender, it's crucial;
  • which criteria for the selection of psychometric tools? 
  • wink at the fear of being stigmatized or discriminated as HCW or public utility worker, simply if they result SARS-CoV-2 positive or get CoViD-19, do vaccination counseling or simply will get vaccinated. This is a limitation that you may consider and deal with;
  • Discussion must be improved;
  • Where are Conclusions? What are the possible repercussions? What suggestions to give to the health policy maker? Define a clear "take home message" from your perspective and address a conclusion section. You need conclusions.
  • Please state in the conclusion if you will re-contact participants to retake the questionnaire after the pandemic or after being vaccinated;

Please update these gaps referring to the following references:

  • Irigoyen-Camacho, M.E.; Velazquez-Alva, M.C.; Zepeda-Zepeda, M.A.; Cabrer-Rosales, M.F.; Lazarevich, I.; Castaño-Seiquer, A. Effect of Income Level and Perception of Susceptibility and Severity of COVID-19 on Stay-at-Home Preventive Behavior in a Group of Older Adults in Mexico City. Int. J. Environ. Res. Public Health 2020, 17, 7418
  • Baldassarre, A.; Giorgi, G.; Alessio, F.; Lulli, L.G.; Arcangeli, G.; Mucci, N. Stigma and Discrimination (SAD) at the Time of the SARS-CoV-2 Pandemic. Int. J. Environ. Res. Public Health 2020, 17, 6341
  • Sarah Dryhurst, Claudia R. Schneider, John Kerr, Alexandra L. J. Freeman, Gabriel Recchia, Anne Marthe van der Bles, David Spiegelhalter & Sander van der Linden (2020) Risk perceptions of COVID-19 around the world, Journal of Risk Research, DOI: 10.1080/13669877.2020.1758193
  • Wise, T., et al. (2020) Changes in risk perception and self-reported protective behaviour during the first week of the COVID-19 pandemic in the United States. Royal Society Open Science. doi.org/10.1098/rsos.200742

Last but not least:

  • what about vaccine hesitancy in this scenario? deal with it, even because CoViD-19 vaccines are now available all over the globe (please also refer to https://www.who.int/news-room/spotlight/ten-threats-to-global-health-in-2019 )